# Synergistic Effects of Ladder and Cage Structured Phosphorus-Containing POSS with Tetrabutyl Titanate on Flame Retardancy of Vinyl Epoxy Resins

**DOI:** 10.3390/polym13091363

**Published:** 2021-04-22

**Authors:** Xu Han, Xiaohua Zhang, Ying Guo, Xianyuan Liu, Xiaojuan Zhao, Heng Zhou, Songli Zhang, Tong Zhao

**Affiliations:** 1School of Materials Science and Engineering, Jiangsu University, Zhenjiang 212013, China; hanxu960131@163.com; 2Key Laboratory of Science and Technology on High-Tech Polymer Materials, Institute of Chemistry, Chinese Academy of Sciences, Beijing 100190, China; gsummer@iccas.ac.cn (Y.G.); lxyuan323@163.com (X.L.); tzhao@iccas.ac.cn (T.Z.); 3School of Materials Science and Engineering, Beijing University of Chemical Technology, Beijing 100029, China; zhangxiaohua1114@163.com

**Keywords:** vinyl epoxy resin, flame retardant, thermal stability, mechanical properties

## Abstract

The cage and ladder structured phosphorus-containing polyhedral oligomeric silsesquioxanes (DOPO-POSS) have been synthesized through the hydrolytic condensation of 9,10-dihydro-9-oxa-10-phosphenanthrene-10-oxide (DOPO)-vinyl triethoxysilane (VTES). The unique ladder and cage–ladder structured components in DOPO-POSS endowed it with good solubility in vinyl epoxy resin (VE), and it was used with tetrabutyl titanate (TBT) to construct a phosphorus-silicon-titanium synergy system for the flame retardation of VE. Thermal stabilities, mechanical properties, and flame retardancy of the resultant VE composites were investigated by thermal gravimetric analysis (TGA), dynamic mechanical analysis (DMA), three-point bending tests, limiting oxygen index (LOI) measurement, and cone calorimetry. The experimental results showed that with the addition of only 4 wt% DOPO-POSS and 0.5 wt% TBT, the limiting oxygen index value (LOI) increased from 19.5 of pure VE to 24.2. With the addition of DOPO-POSS and TBT, the peak heat release rate (PHRR), total heat release (THR), smoke production rate (SPR), and total smoke production (TSP) were decreased significantly compared to VE-0. In addition, the VE composites showed improved thermal stabilities and mechanical properties comparable to that of the VE-0. The investigations on pyrolysis volatiles of cured VE further revealed that DOPO-POSS and TBT exerted flame retardant effects in gas phase. The results of char residue of the VE composites by SEM and XPS showed that TBT and DOPO-POSS can accelerate the char formation during the combustion, forming an interior char layer with the honeycomb cavity structure and dense exterior char layer, making the char strong with the formation of Si-O-Ti and Ti-O-P structures.

## 1. Introduction

Vinyl ester resin (VE) are generally prepared by the reaction of unsaturated carboxylic acid and epoxy resin. The unique chemical structures endow them with the outstanding characteristics of epoxy resins and the advantages of unsaturated polyester resins [1,2]. Due to its excellent adhesion, mechanical properties, chemical corrosion resistance and easy processability, VE has been widely used in various applications, such as composites, anticorrosion pipelines, adhesives, automotive, etc [3,4]. However, VE is easy to burn, which greatly limits its application in aviation, shipping, and other special fields [5,6,7]. Therefore, the preparation of efficient flame-retardant vinyl epoxy resin has become a hot topic of academic research [8,9,10].

In the past decades, halogenated compounds have been widely used to develop vinyl epoxy resins with excellent flame retardancy. However, with the increasing awareness on environmental protection, the use of halogenated flame retardants in many applications is limited due to the release of corrosive or toxic gases during combustion. Phosphorous flame retardants have attracted more and more attention due to their advantages of high efficiency, smokelessness, low toxicity, and environmental friendliness. In particular, DOPO and its derivatives are widely used to improve the flame retardancy of epoxy resins due to their high thermal stability and flame-retardant efficiency [11,12,13,14,15,16,17,18]. For example, with the addition of 15 wt% 9,10-dihydro-9-oxa-10-phosphaphenanthrene-10-oxide (DOPO) derivative (PN-DOPO) into glass fiber-reinforced polyamide 6T, a V-0 rating, and LOI value of 28.9% was achieved [14].

Silicone flame retardant is a kind of non-toxic, green flame retardant, which produces ceramic phase similar to SiO_2_ during the combustion to improve flame retardancy [19]. Polyhedral oligomeric silsesquioxanes (POSS) are a new type of hybrid silicon flame retardant, which exert astonishing effect on some fire properties of different polymers [20,21,22,23]. For example, Svetlichnyi et al. [20] reported that by covalent bonding of nanosized octahedral silsesquioxanes particles containing reactive glycidyl group to polyamidoimides containing a carboxy group in the pendant chain, new polymeric nanocomposites were prepared, TGA data showed that 5 wt% weight loss temperature of the nanocomposites increased sharply both in an inert medium and air. It has been reported that by adding both POSS and DOPO into epoxy resins, the flame retardant efficiency was increased significantly due to the synergistic effect between phosphorus and silicon [24,25,26]. With the addition of 20 wt% POSS-bisDOPO, LOI value of the epoxy composites could reach up to 34.5% [26]. Marciniec et al. [27] synthesized two phosphorus-containing cage-like silsesquioxane derivatives (4P4GS and 8PS) and used them as reactive or additive flame retardants for epoxy resin. Yang et al. [28,29,30,31,32,33,34,35] synthesized different kinds of cage phosphorus containing polyhedral oligomeric silsesquioxanes through hydrolytic condensation and addition reaction, such as DOPO-POSS, DPP-POSS, and DPOP-POSS. The representative flame retardant DOPO-POSS exhibited high-efficient flame retardancy in epoxy, polycarbonate and polycarbonate/acrylonitrile-butadiene-styrene resins, and the flame retardant mechanisms of DOPO-POSS in epoxy resins has been systematically investigated. However, POSS often has poor compatibility with polymer matrix, which impairs the mechanical property of the composites [36,37]. Fina [36] reported that the cage/ladder structure instead of cage structure polyhedral silsesquioxanes could reduce the interaction strength between the cages of silsesquioxanes, thus the cage/ladder structure polyhedral silsesquioxanes could have better dispersion in polymer matrix.

In addition, it has been reported that the introduction of metals into polymer matrix can also improve flame retardancy of the composites. Zhang et al. [38] found that flame retardant efficiency of shape-stabilized phase change material (FSPCMs) can be improved by the addition of iron. Chen et al. [39] revealed that Fe_2_O_3_ could increase smoke suppression efficiency and thermal degradation temperature of silicone rubber composites. Zeng et al. [40] reported that the tetrabutyl titanate (TBT) could be used as co-additive with POSS-bisDOPO to construct a phosphorous-silicon-titanium synergy system in the flame retardancy of epoxy composites, the titanium could be activated under the heat and play a role of catalyst which can accelerate the formation of char. Gao et al. [41] investigated the synergistic flame retardance of APP, PEPA and MoO3 on the flame retardancy of vinyl ester resins, apparent synergistic effect among APP, PEPA and MoO_3_ could be proven by the results of LOI and UL-94 tests, the LOI and UL-94 result of the vinyl ester resin with 10 wt% APP, 10 wt% PEPA, and 5 wt% MoO_3_ were 31.0 and V-0. And the interaction of the three additives through chemical reaction can heighten the thermal stability and strength of the char. Zhang et al. [42] reported that CaCO_3_ and APP showed effective synergistic action on decreasing the HRR and smoke release rate of vinyl ester resin. With the addition of 5 wt% CaCO_3_ and 20 wt% APP, the LOI value and UL-94 result of the VE composites were 28.6 and V-0, which is attributed to the formation of continuous and dense char layer.

In this work, we have synthesized ladder and cage structured phosphorus-containing polyhedral oligomeric silsesquioxanes and named them DOPO-POSS, then DOPO-POSS were used as co-additive with TBT to construct multi-component synergistic flame retardant system containing phosphorus, silicon and titanium. The cage and ladder structures of DOPO-POSS are conductive to their dispersion in vinyl epoxy resin, and titanium can catalyze the formation of char. The effects of DOPO-POSS and TBT on the thermal stabilities, mechanical properties, and flame retardancy of the VE composites were investigated. The flame-retardant mechanism of the prepared VE composites was also revealed.

## 2. Experimental

### 2.1. Materials

9,10-dihydro-9-oxa-10-phosphaphenanthrene-10-oxide (DOPO) and tetrabutyl titanate (TBT) were purchased from Aladdin Co., Ltd. (Shanghai, China). Triethoxyvinylsilane (VTES) and 2,2′-azobis (2-methylpropionitrile) (AIBN, 99.5%) was purchased from Ron chemical reagent network. MFE-711 epoxy vinyl ester resin (VE), methyl ethyl ketone peroxide (LPT-IN), cobalt isooctanoate (P002) were provided by Hua Chang polymer Co., Ltd., East China University of science and technology. Concentrated hydrochloric acid (36.5%, HCl), N, N-dimethylformamide (DMF) and deionized water were obtained from Beijing Chemical Reagent Factory. All chemicals were used as received.

### 2.2. Synthesis of DOPO-VTES and DOPO-POSS

As shown in Scheme 1, Firstly, DOPO-VTES was synthesized through the addition reaction between P-H groups of DOPO and C=C groups of VTES [32], then DOPO-POSS was synthesized from the hydrolytic condensation reaction of DOPO-VTES. The concentration of monomer, nature of the solvent and content of catalyst can all affect the structure of products [43,44,45,46]. By choosing DMF as solvent and adjusting the concentration of monomer, we expect to obtain DOPO-POSS with cage, ladder, and cage–ladder structured components.

#### 2.2.1. Synthesis of DOPO-VTES

To a 250 mL three-necked flask equipped with a mechanical stirrer, a reflux condenser, a thermometer and a nitrogen inlet, DOPO (32.4 g, 0.15 mol), VTES (28.5 g, 0.15 mol) and AIBN (1.476 g, 0.009 mol) were added and stirred gently until temperature was increased to 80 °C. Then, the reaction mixture was stirred at 80 °C for 6 h to get a light-yellow viscous liquid (DOPO-VTES) (56.86 g, yield: 93.4%). FTIR (KBr, cm^−1^): 3065 cm^−1^ (Ar-H); 2975, 2926, 2892 cm^−1^ (C-H); 1478 cm^−1^ (P-Ph); 1207 cm^−1^ (P=O); 1079 cm^−1^ (Si-O); 910 and 752 cm^−1^ (P-O-Ph). ^1^H-NMR (DMSO-d_6_, ppm): 0.50–0.75 (2H, -Si-CH_2_), 0.85–1.26 (9H, CH_3_), 1.96–2.11 (2H, -P-CH_2_), 3.58–3.88 (6H, -O-CH_2_), 7.19–8.51 (8H, Ar-H).

#### 2.2.2. Synthesis of DOPO-POSS

DOPO-VTES (56.86 g, 0.14 mol), deionized water (30 mL), HCl (10 mL) and DMF (150 mL) were added into a 250 mL three-necked flask equipped with a mechanical stirrer, a thermometer and a nitrogen inlet, the mixture was stirred at 80 °C for 30 h, and then the mixture was poured slowly into 500 mL deionized water to yield a white precipitate. The white precipitate was washed three times and suction filtration. After being vacuum dried at 150 °C for 6 h, a white solid (DOPO-POSS) was obtained (50.43 g, yield: 88.7%). FTIR (KBr, cm^−1^): 3472 cm^−1^ (Si-OH); 3066 cm^−1^ (Ar-H); 2908 cm^−1^ (-P-CH_2_-CH_2_); 1478 cm^−1^ (P-Ph); 1230 cm^−1^ (P=O); 1116 and 1080 cm^−1^ (Si-O-Si); 912 and 752 cm^−1^ (P-O-Ph). ^1^H-NMR (DMSO-d_6_, ppm): 0.50–0.95 (2H, -Si-CH_2_), 1.96–2.25 (2H, -P-CH_2_), 6.70–8.30 (8H, Ar-H). ^29^Si-NMR (CDCl_3_-d_6_, ppm), -55 (Si-OH), -65 (Si atoms of completely condensation).

### 2.3. Preparation of the VE Composites

The formulations of the VE composites are listed in Table 1. Firstly, DOPO-POSS was dispersed in MFE-711 resin by mechanical stirring at 70 °C for 30 min to get a clear liquid. After the mixture was cooled to room temperature, the curing agent (LPT-IN) and accelerator (P002) were added and stirred at room temperature for 10 min. After degassing under reduced pressure, the mixture was poured into the steel mold and cured at 120 °C for 2 h in a convection oven. After curing, all samples were cooled to room temperature. The schematic diagram of process as shown in Scheme 2.

The samples containing TBT were prepared by the same method, except that different amounts of TBT were added with the addition of 4 wt% DOPO-POSS. In addition, a group of pure VE was prepared as control group.

### 2.4. Characterization

FTIR spectra were recorded in the range of 4000~400 cm^−1^ on Bruker tensor 27 FTIR instruments (Bruker, Germany). ^1^H-NMR and ^29^Si-NMR spectra were obtained with Bruker AVANCE 400 MHZ NMR instrument (Bruker, Germany) using CDCl_3_ as the solvent and tetramethylsilane (TMS) as the internal standard. Wide-angle X-ray diffraction (XRD) measurements (Bruker, Germany) were performed at room temperature on EMPYREAN X-ray diffractometer at 40 kv and 40 mA with CuKα radiation (λ = 0.1541 nm), scanning range 4~40°. MALDI-TOF test on Bruker ultraflextreme MALDI-TOF/TOF.

The limiting oxygen index (LOI) values were evaluated on a JF-3 Oxygen index instrument according to GB/T 2406.2-2009. The size of the samples was 100 × 10 × 3 mm^3^, fifteen samples were taken from each group. Cone calorimetry measurements were performed on a FTT cone calorimetry according to the ISO 5660 standard under an external heat flux of 35 KWm^−2^. The size of the VE thermosets was 100 × 100 × 10 mm^3^ and three specimens were tested for every sample.

Thermo gravimetric analysis (TGA) was determined on a STA449F5 thermal analyzer (NETZSCH, Germany) under N_2_ and air atmosphere at 10 °C/min from 25 to 800 °C. Dynamic mechanical analysis (DMA) was measured on TA Q800 with the following conditions: frequency 1 Hz, heating rate 3 °C/min, temperature range of 30~150 °C. Three-point bending test was investigated on an AGS-X electronic testing machine. The size of the VE thermosets was 80 × 10 × 4 mm^3^.

Scanning electron microscope (SEM) was recorded with a Hitachi S-4800(Hitachi, Japan) at an acceleration voltage of 10 KV. Prior to SEM measurements, the surfaces were coated with thin layers of gold of about 100 Å. Raman spectroscopy was determined on LABRAM HR Evolution to further investigate the residual char samples after cone calorimetry test. X-ray photoelectron spectroscopy (XPS) measurement was performed using an ESCALAB250XI instrument (Thermo Fisher Scientific, America). The obtained data were calibrated by C 1s standard peak and analyzed by PEAK XPS software.

TGA was coupled with FTIR (Bruker tensor 27), and the measurements were carried out in air atmosphere at 10 °C/min from 40 to 600 °C

Pyrolysis-gas chromatography/mass spectrometry (Py-GC/MS) analysis was carried out with Exactive GC Orbitrap GC-MS (Thermo Fisher Scientific, America). High temperature cracker: CDS PYROPROBE 6200; Gas chromatograph TRACE 1310; Mass spectrometer equipment ISQ 7000. The temperature of GC/MS interface was 300 °C and the pyrolysis temperature was 900 °C.

## 3. Results and Discussion

### 3.1. Characterization of DOPO-VTES and DOPO-POSS

Appendix A shows the FTIR spectra of DOPO, VTES and DOPO-VTES. As shown in Appendix A, the characteristic absorption peak at 3066 cm^−1^ is assigned to C-H stretching vibration of the aromatic ring. The absorption bands at 2975 cm^−1^, 2926 cm^−1^ and 2892 cm^−1^ from the C-H stretching vibration of the alkyl group are also detected. The absorption bands at 1478 cm^−1^, 1207 cm^−1^ and 910 cm^−1^ are attributed to P-Ph, P=O, P-O-C groups. In addition, the absorption bands at 2437 cm^−1^ corresponding to the P-H characteristic peak disappeared, indicating the successful addition reaction between the P-H groups of DOPO and C=C groups of VTES.

The ^1^H-NMR of DOPO, VTES, and DOPO-VTES are shown in Appendix A. It can be seen that the characteristic peak at 8.90 ppm assigned to the protons in P-H of DOPO (Appendix A) and the signals at 5.80–5.95 and 6.07–6.14 ppm assigned to the protons of -CH=CH_2_ of VETS (Appendix A) disappeared in DOPO-VTES (Appendix A), confirming the successful addition reaction between DOPO and VTES.

Appendix A shows the FTIR spectra of DOPO-VTES and DOPO-POSS. In Appendix A, the absorption of -O-CH_2_-CH_3_ group of DOPO-VTES appearing at around 2975, 2926, and 2892 cm^−1^ greatly decreased in Appendix A, indicating the successful hydrolysis condensation of the ethoxy group. The absorption bands at 3472 cm^−1^ were attributed to Si-OH stretching vibration, and the peak at 2908 cm^−1^ was attributed to the -P-CH_2_-CH_2_ group. In addition, the peaks at 1000–1200 cm^−1^ were assigned to Si-O-Si absorptions, in particular, the 1116 cm^−1^ band was assigned to the symmetrical cage structure, while the 1080 cm^−1^ band was assigned to the random structure [47], indicating that the ladder and cage structured DOPO-POSS has been successfully synthesized.

Appendix A shows the ^1^H-NMR spectra of DOPO-POSS. It can be seen that the signals at 0.86–1.26 and 3.58–3.88 ppm assigned to the -CH_3_ and -O-CH_2_ of DOPO-VTES in Appendix A disappeared in Appendix A, revealing the complete hydrolysis condensation of DOPO-POSS. Moreover, the signals at 0.5–0.95 and 1.96–2.25 ppm were assigned to the protons of Si-CH_2_- and P-CH_2_-, respectively.

The ^29^Si-NMR spectra of DOPO-POSS are showed in Appendix A. The signals at –65 ppm were ascribed to Si atoms of complete condensation [48], and the signals at –55 ppm were assigned to the Si-OH in DOPO-POSS.

Appendix A shows the XRD profile of DOPO-POSS. It can be seen that DOPO-POSS exhibits three peaks at 2θ = 5.46 ^o^, 13.18 ^o^ and 20.68 ^o^ corresponding to repeat distances of approximately 16.1 Å, 6.71 Å, and 4.29 Å, indicating the amorphous nature of the polymer. The narrow peak at 2θ = 5.46 ^o^ corresponds to the intermolecular distance and the wide peaks at 2θ = 13.18 ^o^ and 2θ = 20.68 ^o^ corresponds to the intramolecular distance. Compared with the reported polyphenylsiloxane (PPSQ) which has the chain-to-chain distance of 12.5 Å and the intramolecular distance of 4.6 Å [15,49], DOPO-POSS has two kinds of intramolecular distances (6.71 Å and 4.29 Å), which may be attributed to the cage and ladder structure of DOPO-POSS.

The MALDI-TOF mass spectrum of DOPO-POSS is shown in Appendix A. In the spectrum, peaks in the m/z range 2000–3000 were detected, and the assignments of the major components are shown in Table 2. It can be seen from Table 2 that the differences in m/z between the two adjacent ladder structure peaks were 304 (RSiO_2_H) and 286 (RSiO), and two adjacent cage and cage–ladder structure were 304 (RSiO_2_H), respectively. The chemical structures of the representative cage and ladder structured components are illustrated in Figure 1. It can be seen that DOPO-POSS are composed of cage, ladder and cage–ladder structured components, and most of the components are ladder structure [50], which is consistent with the results of FTIR and XRD. The ladder and cage–ladder structures are beneficial to reduce the crystallinity of POSS, which improves the compatibility between DOPO-POSS and vinyl epoxy resin [36]. The presence of Si-OH groups also increases the affinity of synthesized DOPO-POSS to polar VE resin.

### 3.2. Morphologies and Mechanical Properties of the VE Composites

Figure 2 shows the SEM micrographs of the fractured surfaces of the VE composites. It can be seen that at low loadings (≤4 wt%), no obvious particles or agglomeration are observed from the fractured surfaces of both VE-0, VE-1, and VE-2, indicating that the DOPO-POSS can be well dispersed in the VE matrix. At high loadings (5 wt%), aggregates of micrometer size could be observed on the fractured surface of VE-3, which is attributed to the separation of DOPO-POSS from VE matrix. The good solubility of DOPO-POSS in VE matrix was due to the presence of the ladder and cage–ladder structure components.

Figure 3 shows the σ-ε curves of the VE composites tested by three-point bending and the results are shown in Table 3. It can be seen that flexural strength of VE composites first increased and then decreased with the increase of DOPO-POSS in VE matrix. Generally speaking, the fracture strain of materials depends on both stiffness and toughness [35]. The improved fracture strain of VE composites with the addition DOPO-POSS may be attributed to the increased content of rigid benzene ring structure and cage structure of DOPO-POSS which greatly inhibit the movement of polymer molecular chain in thermoset networks [51,52,53]. For the control sample (VE-0), the flexural strength and modulus of VE-0 were 76.54 MPa and 2.39 GPa. As for VE-2 and VE-2-2, flexural strength of the composites increased to 77.46 and 79.17 MPa. However, as for VE-3, flexural strength decreased to 71.77 MPa because of the separation of DOPO-POSS from VE matrix, which is consistent with the results of SEM.

### 3.3. Thermal Stability

TGA and DTG curves of DOPO, DOPO-POSS and the VE composites under nitrogen and air atmospheres are shown in Figure 4. And the relevant thermal decomposition data, including T_5%_ which is defined as the temperature at 5 wt% weight loss, T_max_ which is defined as the temperature at maximum weight loss rate, and the char residues at 800 ℃ are summarized in Table 4.

As shown in Figure 4a,b, under the nitrogen atmosphere, the T_5_% and T_max_ of DOPO-POSS and DOPO are 384.0 and 479.4, 267.0, and 332.3 °C, respectively, and their char residues at 800 °C are 41.1% and 3.92%. Figure 4c,d show the results under the air atmosphere, the T_5_% and T_max_ of DOPO-POSS and DOPO are 304.3 and 483.7, 234.7 and 271.7 °C, respectively, and their char residues at 800 °C are 37.71% and 3.07%. It can be observed that a two-stage degradation process occurred in DOPO, the first degradation process occurred at around 267 °C, which may be attributed to decomposition of the weak P-O-C bonds [9,16,54]. The second degradation process occurred at around 340 °C, which may be attributed to the degradation of aromatic ring [16,55]. Moreover, DOPO-POSS exhibited high thermal stability because it has SiO_2_ cage core and rigid Si-O-Si structure could change the thermal decomposition process through the formation of thermal stable SiO_2_ ceramic phase [30,34,56].

The T_5%_ and the char residues of VE-0 were around 321.2 ℃ and 7.07%. With the addition of DOPO-POSS into VE matrix, the T_5%_ and the char residues of VE composites increased significantly. Compared to VE-2, with the addition of 0.5 wt% TBT, the char residues of VE-2-2 increased from 9.35% to 14.38%. This may be due to titanium-containing TBT having acted as a catalyst to promote the formation of char at high temperature [18].

As shown in Figure 4c,d, under the air atmosphere, it can be observed that a three-stage degradation process occurred in all samples, the first and second decomposition process occurred at around 360 °C, and reached T_max_ at about 410 °C, which may be attributed to the degradation of aromatic ring and alkyl chain [16]. The third decomposition process appeared at around 520 °C, mainly due to the further thermal oxidative degradation of the unstable char layer formed by aromatic ring and alkyl chain.

Appendix A shows the curves of the storage modulus and tan δ of the pure VE and the VE composites. The glass transition temperature (Tg) of the samples is determined from the peak temperature of tan δ curves. The storage modulus at 40 (E′ _40_) and 150 °C (E′ _150_), tan δ and Tg are summarized in Appendix A. In general, T_g_ is determined by the cross-linking density of the resin, the rigidity of the chain structure and the segmental motion freedom, etc. As can be seen from Appendix A, the E′ _150_ of all the VE composites are lower than that of VE-0, indicating that the crosslinking density of the VE composites are reduced [57]. On the one hand, the large volume of rigid benzene ring structure and cage structure of DOPO-POSS can greatly inhibit the movement of polymer molecular chain, which increased T_g_; on the other hand, the addition of bulky DOPO-POSS decreased the cross-linking density of VE resins, which decreased T_g_. Compared to VE-0, the glass transition temperature of VE-2 and VE-2-2 were slightly increased due to the good dispersion of DOPO-POSS in VE matrix. In addition, the glass transition temperature of VE-3 decreased compared to VE-0, which was attributed to the separation of DOPO-POSS from VE matrix at high loadings, which is consistent with the results of SEM.

### 3.4. Flame Retardant Properties of the VE Composites

LOI test was used to evaluate the flame retardancy of the VE composites, and the detailed data are shown in Table 5. With the increase of DOPO-POSS, the LOI values of the composites increased from 19.5 to 22.1. It is interesting that with the introduction of DOPO-POSS and TBT, the LOI values of VE composites were improved significantly. When 4 wt% DOPO-POSS and 0.5 wt% TBT were incorporated into VE, the LOI value reached 24.2, which was improved by 24.1% compared with that of the pure VE. The results revealed that TBT played a key role in increasing the LOI value of the vinyl epoxy resin [38,39,40], and phosphorus, silicon, and titanium showed a good synergistic effect in the flame retardancy of VE composites.

The combustion behavior of the polymer was further investigated by cone calorimetry. Figure 5 and Figure 6 show the total heat release (THR), the heat release rate (HRR) curves of the VE composites, and the key combustion parameters are summarized in Table 6. As shown in Figure 5a, after cone calorimetry test, the amount of residual char of the VE composites increased gradually in the order of VE-0, VE-2 and VE-2-2. From Figure 5b,c, it can be seen that pure VE burns quickly after ignition and the peak heat release rate (PHRR) and total heat release (THR) were 616.7 Kwm^−2^ and 395.8 MJm^−2^, while those of VE-2 were 299.9 Kwm^−2^ and 267.1 MJm^−2^, which were reduced by 51.4% and 32.5% compared with those of the pure VE. As for VE-2-2, the PHRR decreased to 264.5 Kwm^−2^ while THR increased slightly to 289.2 MJm^−2^. As can be seen from Figure 6, smoke production rate (SPR), total smoke production (TSP), average of CO_2_ Yield (av-CO_2_Y), average of CO Yield (av-COY) of VE-2 and VE-2-2 are all lower than those of VE-0. The peak of SPR, TSP, av-CO_2_Y and av-COY of pure VE were 0.127 m^2^s^−1^, 85.15 m^2^, 2.15 kgkg^−1^ and 0.06 kgkg^−1^, while those of the VE-2-2 decreased significantly to 0.104 m^2^s^−1^, 73.45 m^2^, 0.69 kgkg^−1^ and 0.002 kgkg^−1^ by 18.1%, 13.7%, 67.9%, and 96.7%, respectively. This illustrated that DOPO-POSS and TBT have a good synergistic effect, could efficiently facilitate the generation of compact residual char layer that prevented further degtadation of the matrix into organic volatiles or gases and inhibited the burning effectively [52].

### 3.5. Flame-Retardant Mechanism

#### 3.5.1. Condensed Phase Analysis

Figure 7 shows the photographs of the VE composites after heated at 450 ℃ for 30 min in the muffle furnace under the air atmosphere. It can be seen that the char residue of the pure VE have been completely destroyed to thin and fragile fragments. With the addition of 4 wt% DOPO-POSS, the char residues of VE-2 and VE-2-2 retained intact [58].

The morphology of char layers after cone calorimetry testing was further investigated by SEM. As shown in Figure 8a,a’, the exterior and interior char of VE-0 is porous, thin and brittle. In contrast, in the Figure 8b,b’,c,c’, after adding DOPO-POSS, the exterior residual char became dense and continuous, and the interior char layer had the characteristic of honeycomb cavity structure, which could hamper the heat flow and mass transport [59].

Figure 9 shows the Raman spectrum of residual char after cone calorimetry. The characteristic peaks of 1350 cm^−1^ (D band) and 1590 cm^−1^ (G band) are disordered and ordered char, respectively [60]. The compactness of the char layer can be measured by the ratio of I_D_/I_G_ peak intensity. The I_D_/I_G_ value decreases in the order of VE-0 (0.85) > VE-2 (0.76) > VE-2-2 (0.74), indicating that the incorporation of DOPO-POSS increased the compactness of the residual chars, which is in accordance with the results from SEM. With the addition of TBT, more compact char residue was formed in sample VE-2-2 [39,40], which could provide better physical barrier effect, improving the flame retardancy of the VE composites.

To further explore the flame retardant mechanism in condensed phase, the exterior and interior residual chars of VE-2 and VE-2-2 are studied by EDX analysis (Figure 10). It can be seen that for VE-2 and VE-2-2, the main elements in the char layers are carbon and oxygen, and a small amount of P and Si. Compared with VE-2, new signals belonging to Ti element appeared in VE-2-2, and the content of P and Si elements increased evidently. It can be inferred that the addition of Ti element into the VE composites was beneficial to the formation of dense and stable char layer, and different Ti containing compounds may be formed during combustion [53,61,62], which was helpful to improve the fire resistance.

The exterior residual char VE-2 and VE-2-2 were studied by XPS to further investigate the mechanism of char formation. Figure 11a illustrated the XPS spectra of VE-2 and VE-2-2, and the Si2p, P2p, C1s and O1s spectra of VE-2 are shown in Appendix A. As shown in the Si2p spectrum of VE-2-2 (Figure 11b), the characteristic binding energy peaks at 102.5 eV, 102.8 eV, 103.5 eV, and 104.4 eV were attributed to the Si-O-Ti, Si-C, Si-O/Si-O_2_, and -P(=O)O-Si [63,64,65,66], respectively. In the P2p spectrum of VE-2-2 (Figure 11c), three peaks at 133.3 eV, 133.9 eV, and 134.4 eV were attributed to Ti-O-P, P-O-C and -P(=O)O-Si groups [53], respectively. In the Ti2p spectrum of VE-2-2 (Figure 11d), Ti-O-P (459.4 eV) and Ti-O-Si (466.0 eV) [67,68,69] components can be clearly identified. In the C1s spectrum of VE-2-2 (Figure 11e), there are four peaks at 284.1 eV, 284.7 eV, 286.1 eV, and 288.9 eV, which are attributed to C-Si, C-C, C-O and C=O groups [53,70], respectively. In the O1s spectrum of VE-2-2 (Figure 11f), there are four peaks at 530.9 eV, 531.8 eV, 532.6 eV, and 533.5 eV, which are attributed to Ti-O, C=O, P-C-O and P-O-P groups [63,65], respectively. The above results indicated that the addition of TBT was conducive to the formation of different Ti containing compounds in the char, which is helpful to improve flame retardancy of VE composites.

#### 3.5.2. Pyrolysis Behaviors of the VE Composites

Figure 12 shows the 3D TG-FTIR and FTIR spectra of the gas phase at different temperatures of pyrolysis of VE-0 and VE-2. As shown in Figure 12a,b, the pyrolysis products of VE-0 and VE-2 are obviously different. The evolved gas components of VE-0 and VE-2 in air at 322, 421 and 560 °C are shown in Figure 12c,d. With the increase of pyrolysis temperature, the number of absorption peaks first increased then decreased. From the evolved gas at 421 °C for VE-2, the bands of -OH and phenol (3900–3500 cm^−1^), aromatic components (3030, 1503 cm^−1^), hydrocarbons (2940 cm^−1^), CO_2_ (2376 cm^−1^), and ether components (1768, 1128 cm^−1^) were identified, which was similar to those of pure VE [71,72,73]. In addition, new absorption peaks of P(O)-OH (831 cm^−1^), P=O (1265 cm^−1^), PO_2_ (1175 cm^−1^), and PO-H (3652 cm^−1^) were assigned the phosphorus-containing compounds [74], indicating that DOPO-POSS exerted flame retardant effect in gas phase.

To further explore the flame retardant mechanism of VE composites in gas phase, Py-GC/MS was adopted at the pyrolysis temperature of 900 ℃. Figure 13 and Figure 14 showed the total ion chromatogram (TIC) and the typical MS spectra of the DOPO-POSS, VE-0, VE-2, and VE-2-2, and the possible structural assignments of the VE-2 are listed in Table 7. It can be seen that the main pyrolysis products of VE-2 are (E)-2-methyl-3-phenylacrylaldehyde (m/z = 146), (2-methylprop-1-en-1-yl) benzene (m/z = 132), (E)-prop-1-en-1-ylbenzene (m/z = 118), styrene (m/z = 104), toluene (m/z = 92), benzene (m/z = 78), cyclopenta-1,3-diene (m/z = 66), cyclobuta-1,3-diene (m/z = 52), and the m/z of fragments at 134, 119, 91, 77, 63 were ascribed to PO_2_· and HPO_2_·free radicals [75,76], which indicated that phosphorus-containing fragments were released into the gas phase during pyrolysis.

To sum up, the mechanism of flame retardancy was as follows: the radical quenching effect of phosphorus in the gas phase; the stable SiO_2_ ceramic phase formed by Si and the synergistic effect between phosphorus and Ti accelerated the formation of residue char, which had the characteristic of honeycomb cavity in the interior layers and compact exterior layers, preventing the heat flow and transfer to improve the flame retardancy of the composites.

## 4. Conclusions

In this work, a ladder and cage structured phosphorus-containing polyhedral oligomeric silsesquioxanes (DOPO-POSS) was synthesized and characterized. The unique cage and ladder structure of DOPO-POSS facilitated its good solubility in the VE composites. DOPO-POSS and TBT was used as flame retardant additives to improve the flame retardancy of cured vinyl epoxy resin. Under the nitrogen atmosphere, T_5%_ increased from 321.2 to 353.3 °C and char residue increased from 7.07 to 14.38% compared to pure VE. With the incorporation 4 wt% DOPO-POSS and 0.5 wt% TBT, the LOI value of the VE composites increased from 19.5 to 24.2, and the PHRR, THR, SPR, and TSP were reduced by 57.1%, 26.9%, 18.1%, and 13.7%, respectively. In addition, the VE composites showed comparable mechanical properties to that of the pure VE. The flame retardant mechanism was mainly due to the radical quenching effect of phosphorus, the formation of stable SiO_2_ ceramic phase, the catalytic char generation of Ti and the char forming of phosphorus. All the results indicated that DOPO-POSS and TBT combination have great potential applications in the future.

## Data Availability

The data presented in this study are available on request from the corresponding author.

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
