# Peer review of "Synergistic Effects of Ladder and Cage Structured Phosphorus-Containing POSS with Tetrabutyl Titanate on Flame Retardancy of Vinyl Epoxy Resins"

_polymers, 2021, doi:10.3390/polym13091363_

Round 1
Reviewer 1 Report
Each experiment/project design should be preceded by an intensive literature review. The various DOPO-POSS systems were synthesized many years ago and extensively studied as a flame-retardant for resins, carbonates, and other synthetic materials. Unfortunately, the authors omitted all of these papers. DOPO-POSS compounds were obtained through both, hydrolytic condensation and addition reaction. Moreover, the functionalization of well-defined silsesquioxanes with DOPO is also commonly known and was previously reported. With regard to the above, the novelty of this paper is very low. I don't recommend this paper for publication.
exemplary DOPO-POSS references:
- DOI: 10.1016/j.polymdegradstab.2011.09.016
- doi: 10.1016/j.polymdegradstab.2012.05.020
- doi: 10.1016/j.polymdegradstab.2013.11.015
- doi: 10.1016/j.polymdegradstab.2011.07.014
- doi: 10.1016/j.compscitech.2016.02.026
- doi: 10.1002/app.35203
- doi: 10.3390/ma13235373
- DOI: 10.1002/pat.1929
Author Response
Thank you so much for the valuable comments. Please see the attachment. Attached are the revised manuscript and author's Reply. In the revised manuscript, we have maked with a grey background.

Reviewer 2 Report
This study reports on synergistic effects of ladder and cage structured phosphorus-containing POSS with tetrabutyl titanate on flame retardancy of vinyl epoxy resins. And a series of experimental results have been presented to support it. After carefully reading it, I suggest to consider the following points.
1. I suggest to consider to rewrite the Abstract, please do not present the work details, please consider to explain the working principle of synergistic effect of two filler on polymer matrix. And how about the constitutive relationship between the synergistic effect and property improvements, especially for the flame retardancy. There are several previous works have been conducted on it, and is useful for the reference, i.e., (1) Yong Tang, Jinfeng Zhuge, Jeremy Lawrence, James Mckee, Jihua Gou, Christopher Ibeh and Yuan Hu. Flame retardancy of carbon nanofibre/intumescent hybrid paper based fibre reinforced polymer composites. Polymer Degradation and Stability. Volume 96, Issue 5, May 2011, Pages 760-770. (2) Haibao Lu, Yongtao Yao, Wei Min Huang, Jinsong Leng, David Hui. Significantly improving infrared light-induced shape recovery behavior of shape memory polymeric nanocomposite via a synergistic effect of carbon nanotube and boron nitride. Composites Part B: Engineering. 2014, 62: 256-261. (3) Echeverria, Claudia A.; Ozkan, Jerome; Pahlevani, Farshid; Willcox, Mark; Sahajwalla, Veena. Multifunctional marine bio-additive with synergistic effect for non-toxic flame-retardancy and anti-microbial performance. SUSTAINABLE MATERIALS AND TECHNOLOGIES. 2020, 25, e00199.
2. “successfully” in Introduction section is suggested to be removed. It is not an objective word. Furthermore, it is not necessary.
3. “3.1. Synthesis and Characterization of DOPO-VTES and DOPO-POSS”, the synthesis content is suggested to be moved to the second section, not in this “3. Results and Discussion”.
4. “σ-ε” curves, but not the “ε-σ”. The breaking points should be marked in the figure, not in the table 3.
5. there are so many experimental results have been carried out. The DMA, and some of the Figures 1-7 for DOPO (a), VTES (b) and DOPO-VTES should be presented in supply materials. This study aims on the composite and the flame retardancy, not the filler.
In all, a useful study, and it is worthy to be recommended after revision.
Author Response
Thank you so much for the valuable comments. Please see the attachment. In addition, we have marked with a yellow background In the revised manuscript.

Reviewer 3 Report
Thank you for the opportunity in reviewing the manuscript (polymers-1168253). Interesting results are reported and the manuscript was reviewed for publication in Polymers-MDPI Journal. However, the paper require substantial revision before acceptance and consideration for publication. Few points are -
- What do author mean by ladder and cage structure in title?
- In the introduction, proper literature survey on synergistic effects between binary additives was not performed. Please cite more papers on mainly synergistic effects among the additives especially for flame retardancy of vinyl epoxy resins.
- In section 2.2. and 2.3, please describe the scheme of process, it will be highly appreciated.
- In section 4, #SEM, how SEM were performed. Do the authors do coating prior to the SEM measurements? If yes, which metal was used? Studies on purity of the specimen via SEM-EDX will be highly appreciated. Then, correlate the purity measurements with the properties of composites.
- Please describe and discuss the table 2 (MALDI-TOF data of DOPO-POSS) in result and discussion section.
- In Table-3 (mechanical properties) , please also discuss the values of fracture strain?
- In Figure 11 (TGA), the profiles of DOPO shows two slopes of degradation. Please dicuss them why and their importance in discussion section. Moreover, the thermal stability of DOPO-POSS is highest. Why?
- From Figure 12 and table-5, it can be noted that the glass transition temperature is significantly affected with respect to the different composites. Please discuss it why? Is it is related to polymer crystallinity or other factors?
Author Response
Thank you so much for the valuable comments. Please see the attachment. In addition, we have marked with a green background In the revised manuscript.

Round 2
Reviewer 1 Report
The Authors highlighted the differences between their work and the papers I listed containing a similar approach for flame-retardancy of the organic polymers. I think that the differences should be also mentioned in the manuscript. For example: the Authors of the submitted manuscript as well as Yang et al. obtained DOPO-POSS compounds through hydrolytic condensation of DOPO-VTES, however, the reactions were conducted in different conditions. It should be discussed briefly in the text how the reaction conditions influence the products formation.
- the quality of the 29Si NMR spectrum of DOPO-POSS is poor and doesn't confirm the existence of structures that are postulated by the Authors. The intensity ratio between the peaks and baseline is too low, therefore in my opinion it's very hard to distinguish species present in the isolated product. The experiment should be repeated with significantly more scans.
- moreover "4P4GS, 8PS mentioned in the introduction" were prepared by Marciniec group, not Yang doi: 10.3390/ma13235373
- The Authors presented the following statement "The good solubility of DOPO-POSS in VE matrix was due to the presence of the ladder
and cage-ladder structure components." It seems that is true, furthermore, it could be explained by chemical formulas of ladder-cage structures. Typical perfect cage DOPO-POSS has no free polar groups, on the other side, the structures presented by the Authors contain silanol groups, which are polar, therefore, the presence of SiOH groups increases the affinity of synthesized DOPO-POSS mixture to polar resin. - The good dispersion of the DOPO-POSS in the resin matrix could be also confirmed by EDS experiments, which clearly show the distribution of silicon and phosphorus atoms in the resin.
Author Response
Thank you so much for the valuable comments. Please see the attachment. In this paper, we will marked them with yellow background.

Reviewer 3 Report
Minor comments -
[1] In section 2.3, please describe the schematic diagram of process, it will be highly appreciated.
[2] In Table-3 (mechanical properties) , please also write the values of fracture strain?
[3] From Figure S8 and table S1, it can be noted that the glass transition temperature is significantly affected with respect to the different composites. Please discuss it why? Is it is related to polymer crystallinity or other factors?
Author Response
Thank you so much for the valuable comments. Please see the attachment. In this paper, we marked it with green background.
